# Predictive Modeling of Induction-Hardened Depth Based on the Barkhausen Noise Signal

**DOI:** 10.3390/mi14010097

**Published:** 2022-12-30

**Authors:** Jonas Holmberg, Peter Hammersberg, Per Lundin, Jari Olavison

**Affiliations:** 1RISE Research Institutes of Sweden AB, Argongatan 30, 431 53 Mölndal, Sweden; 2Department of Industrial and Materials Science, Chalmers University of Technology, 412 96 Gothenburg, Sweden; 3Lundin Stress Service AB, Rönningevägen 34, 144 61 Rönninge, Sweden; 4Volvo Group Trucks Operations, Volvovägen 5, 541 87 Skövde, Sweden

**Keywords:** Barkhausen noise, induction hardening, predictive modeling

## Abstract

A non-destructive verification method was explored in this work using the Barkhausen noise (BN) method for induction hardening depth measurements. The motive was to investigate the correlation between the hardness depth, microstructure, and the Barkhausen noise signal for an induction hardening process. Steel samples of grade C45 were induction-hardened to generate different hardness depths. Two sets of samples were produced in two different induction hardening equipment for generating the model and verification. The produced samples were evaluated by BN measurements followed by destructive verification of the material properties. The results show great potential for the several BN parameters, especially the magnetic voltage sweep slope signal, which has strong correlation with the hardening depth to depth of 4.5 mm. These results were further used to develop a multivariate predictive model to assess the hardness depth to 7 mm, which was validated on an additional dataset that was holdout from the model training.

## 1. Introduction

A major challenge in automotive industry when producing heat-treated engine parts is given by acquiring of necessary material properties (e.g., surface hardness, microstructure and hardening depth) within specification limits of a given component to withstand loads during its use. Therefore, the verification of these properties from manufacturing is essential. One of these components are cam shafts for heavy duty vehicles, which is in a great need of a non-destructive characterization (NDC) method to verify the material properties after heat treatment. Currently, this verification is solely done by first-part destructive testing, where manufactured parts are sectioned to smaller pieces and the microstructure and hardness are verified relative the operational window of the process. For cam shafts, this is necessary every time the induction hardening process is reset or when other planned or un-planned interruptions occurs. However, this verification process is very costly due to production stands still during process performance verification.

In the present investigation, the basic relationships for the development of a NDC method for the induction hardening process of cam shafts for heavy duty vehicles was considered. During induction hardening, the surface of the starting material undergoes a rapid heating and quenching, which results in a martensite transformation of the surface layer. This is performed sequentially from one cam lobe to the other, which generates a much harder and wear resistant surface at each sub-section of the part treated. 

The need for non-destructive verification of the heat treatment process is important and prior investigations have shown great potential in the Barkhausen noise (BN) method [1,2] with several potential benefits: higher process capability to lower cost and a more even quality of the surface treated.

The BN method is sensitive to residual stresses, hardness, and microstructure of ferromagnetic materials [3,4,5]. Compressive stresses and high hardness provide a low BN-signal, while tensile stresses and low hardness results in high BN-signal. The BN are the irreversible electromagnetic response of the magnetic domain wall movement in the material surface when a ferromagnetic material is exposed for an external magnetic field [6,7]. The domain wall movement is defined by the domain size, distribution, and the amount of pinning sites as described by the Kittel [8]. These pinning sites are dislocations and line defects in the microstructure, which are directly related to the material hardness, since they aggravate plastic strain. This relationship has been used in different works, i.e., by Tam et al. and Santa-aho et al., who made advancement of the analysis methodology of how to characterize the response signal for sub-surface microstructural characterization for case depth measurements [9,10]. 

Barkhausen noise is well-known of its sensitivity to the impact of the microstructure in the near surface zone while the actual signal depth is not understood. Theoretically, the BN signal depth is only a few tenths of millimeters, but empirical studies have shown potential correlation with larger hardening depths, down to several millimeters [9,10,11]. Advancements have been made by developing new measurement techniques such as the magnetic voltage sweep slope method (MVSS), i.e., shown by Santa-aho et al. [10]. This method performs two voltage sweeps at two different frequencies, one low and one high, and calculates the ratio between the maximum slope is used for each frequency. The maximum slope is an indirect measure of the permeability of the material and by taking the ratio, the signal is normalized. Santa-aho et al. further showed a correlation between the case hardening depth of steel and the MVSS ratio to depths of 3.5 mm [10]. Sorsa et al. suggested development of predictive models to support non-destructively assessment of nitride hardening depth [12]. However, the correlation with greater hardening depths is yet not fully understood, since the analysis depth of the Barkhausen noise signal also is dependent of the so-called skin effect for eddy-current or magnetic fields, as described by Equation (1).
(1)$$\delta =\frac{1}{\sqrt{\pi f\mu \sigma }}=\frac{1}{A}$$
where δ = depth of penetration of the magnetic field, *f* = frequency of the magnetic field, *μ* = magnetic permeability, and *σ* = electrical conductivity. This model describes the theoretical penetration depth of the applied magnetic field assuming a homogeneous material property at a specific constant frequency. 

However, since the BN signal is filtered in a frequency range, the equation for depth analysis, *D(x)*, of the Barkhausen noise signal needs to be modified. Therefore, as suggested in the work of Tiitto et al. [13], a variant of the skin effect formula for a frequency range of the BN signal is given by Equation (2).
(2)$$D\left(x\right)=\frac{\int_{f_{1}}^{f_{2}}g\left(f\right)\exp\left[-A\cdot x\cdot f^{1/2}\right]df}{\int_{f_{1}}^{f_{2}}g\left(f\right)df}$$
where *g(f)* is the form of the amplitude variation as a function of frequency. *g(f)* = 1, for random white noise and *f_1_* and *f_2_* are the frequency limits of the filtered frequency range and x as the distance inside of the material. Applying typical settings for a hardened and tempered steel, conductivity of 10^6^ Ω^−1^·m^−1^, µ_r_ = 200, and frequency range, *f_1_* − *f_2_*, 70–200 kHz, equals to a depth of analysis of 0.1 mm. For a non-hardened, mild steel, the depth of analysis will decrease to approximately 40 µm. Harder steel provides a higher analyzing depth and softer steel provides a lower analyzing depth at the same frequency range.

This is closely related to the BH-hysteresis loop, which is completely different for a soft or hard steel affecting the saturation, the width and slope of the loop as shown by Saquet et al., Jiles, and Swallem [14,15,16]. Induction hardening gives rise to a change in the BN response, as the amount of pinning sites increases [8]. This directly affects the root mean square (RMS) parameter, and generates a secondary peak as shown by Saquet et al., who used this to measure hardness depths to 3.5 mm [15]. In a similar manner, Vaidyanathan et al. utilized this secondary peak to measure the hardening depth down to 3.5 mm [17]. For shallow heat-treated layers, Augustis et al. suggested using the BN Power spectrum, which showed to give very high resolution for layers down to 0.8 mm [18]. 

One plausible reason for the relationship between hardening depth and the MVSS is that the signal utilizes the hardness difference between surface and bulk, which then in principle is a measurement of the permeability of the material. As the MVSS method uses two different magnetizing frequencies, resulting in measurements at two different depths, it can be assumed that the MVSS parameter correlate with the gradient of the hardness. This implies that the analyzing depth of the Barkhausen noise signal does not need to be in the same range as the hardening depth.

The major motive in this investigation has been to further explore the correlation between the hardness depth and the Barkhausen noise signals for induction steels with hardening depths used for cam shafts, in the range of 1.8–7 mm. With this knowledge, a statistical approach has been used to generate valid models for predictive analysis of relevant BN signals.

However, it is of great importance to verify the range of the relevant variation that needs to be detected, in the combination of hardening process and material composition at hand, which can be detected with sufficient precision. The literature show that the BN method has great potential, but the resolution, influenced by the microstructure variation and component geometry on data/decision quality, is not fully understood. This has been the target in this work, realized by evaluation of a new approach utilizing the different BN signals in combination. In a future stage, this will be used as input for process monitoring in combination with, for example, control charts, which combined, ease the need of high measurement system precision [19].

This work was divided into three main parts. Part one focused on developing samples with greatest possible variety of hardening depths. The second part was devoted to characterizing the samples by both non-destructive and destructive analysis in order to establishing clear relationships of the influential aspects of correlation between the hardening depth and the BN response. The third part developed and verified predictive models of the non-destructive measurements of the hardening depth. 

## 2. Methodology

### 2.1. Material

Two sets of samples were developed using two different types of induction hardening equipment. Randomly selected samples from both test sets were used to train a predictive model, while the holdout samples were used for verification of the model. Set A was generated with the greatest possible variation of hardness profiles, while set B was developed within the range of set A with the settings used according to Table 1. Using this approach, the developed model became more generic since it handled different induction heating equipment. Both sample sets were produced using laboratory scale induction hardening equipment, with progressive induction hardening. The material of the samples was steel grade C45 (main alloying elements of Fe: Bal, C: 0.45 wt-%, Mn: 0.71 wt-%, Si: 0.23 wt-%) with a ferritic/pearlitic microstructure as seen if Figure 1. The samples had a diameter of 23 mm and a length of 100 mm.

### 2.2. Heat Treatment

The heat treatment was performed using two types of lab-scale induction hardening (IH) equipment, as seen in Figure 2. Both types of equipment consisted of a circular inductor, constructed of three coil loops for equipment 1 and one single loop for equipment 2. The sample was fixtured in a rotational chuck that rotate the sample during heating and a quench shower, which sprays quenching fluid on the sample after heating. Test set A was heat-treated by altering the scanning speed from 2–8 mm/s, the power from 90–100%, pre-heating time of 4 s, and a concentration of the quenchant of 5%. The equipment used for the test set B offered higher powers up to 150 kW. The samples for set B were produced with scanning speed of 2.5–12 s/mm, the power from 42–55%, and the quenchant concentration from 5–11%.

### 2.3. Sample Characterization

The BN-measurements were performed with a Rollscan 300 from Stresstech Oy, using a shielded sensor with normal bandwidth. Measurements were done using a magnetizing voltage of 6 V and a magnetizing frequency of 80 Hz. The collected BN data were analyzed and calculated as the arithmetic average of three measurements, each consisting of 10 bursts, using the full frequency band. The burst analysis of root mean square (RMS), full width half maximum (FWHM), and peak position was done using a moving average smoothing and a polynomial fit for the peak calculation. The Barkhausen noise were measured using two different approaches: MicroScan software from Stresstech of the conventional BN parameters such as: root mean square value (RMS), peak position, and full width half maximum (FWHM);PCCaseDepth software from Stresstech of the magnetizing voltage sweep slope (MVSS), which measured the ratio from the maximum slope of the sweeps of 200 Hz and 20 Hz. In total, the average slope ratios of four sweep measurements were used.

The surface residual stresses were measured with X-ray diffraction using a Stresstech G2R equipment with Chromium radiation. The (211) diffraction plane was used for measurements located at a diffraction angle of 156.4° and the sin^2^ψ, in modified χ mode, was used with 5 tilt angles in the interval of ±40°. 

The microstructure of all samples was evaluated on polished and nital-etched cross section surfaces of the samples. The hardness was measured with a Qness Q10A+ equipment using Vickers method (1 kg) and the Knoop method (200 g). 

Statistical analyses were done of the measured data to develop models and verification using the software JMP Pro 16.2. A principal component analysis (PCA) approach was selected, as the datasets showed a heavy bi-variate correlation between themselves and the hardening depth.

## 3. Results

### 3.1. Non-Destructive Testing

The Barkhausen noise parameters RMS, FWHM, peak position (Pos), and MVSS (200 Hz/20 Hz) versus the hardening depths (HD) are shown in Figure 3 and Figure 4 with set A to the left and set B to the right. These parameters show different relationship and degree of correlations to the hardening depth. The RMS values were of the same magnitude for hardening depths in the interval 1.8–3.2 mm. At a greater depth, the RMS increased drastically down to 5 mm, but not for the greatest depth, 7 mm. The correlation to the hardness depth is rather S-shaped than linear, as seen in Figure 3A). The FWHM showed a more linear correlation with hardening depth for set A compared to set B, increasing gradually, see grey-colored curves in Figure 3. The MVSS-parameter has a very higher degree of linear correlation to the hardening depth down to 4.5 mm, see Figure 4. In the same Figure, the peak position showed a high correlation for the shallow hardness depth interval for validation set B but not for the deeper interval for training set A. 

The surface residual stresses and diffraction peak broadening, also known as full width half maximum (FWHM), were measured with XRD for the samples as shown in Figure 5. These results showed a great variation of stresses for the different samples ranging from high compressive stresses for samples with low hardening depth (HD) to tensile stresses for the samples with high HD. The surfaces residual stresses showed a linear correlation to HD for set A, as shown by the high R^2^ values of linear trendlines, if the 7 mm sample was considered as an outlier. The FWHM values ranged in the interval 3–7°, showing a decrease with a higher HD. The correlation was s-shaped for set A while set B showed a great variation. 

### 3.2. Destructive Evaluation

The series of heat treatments to generate sets A and B produced a variety of hardness profiles for the different settings of scanning speed, power, and quenchant concentration, as seen in Figure 6. The effective hardening depth (EHD) was determined as the depth where the hardness dropped below 400 HV, in accordance with SAE J423 [20]. The results in Figure 5 show that an increasing scanning speed generated a lower hardness depth and a sharper transition between hardened and non-hardened materials. It could also be observed that decreasing the power from 100% to 90% resulted in a shift towards lower hardness depth for set A. It is further seen that the surface hardness was less influenced of the scanning speed. Test set A show a wide range of hardening depth of 1.7–7.0 mm while set B show a narrow range of 2.4–4.3 mm.

The surface hardness using the Vickers HV1 method in Figure 6 was a too rough measurement and instead the surface hardness was evaluated using the Knoop method with a load of 200 g. This enabled us to measure closer to the surface, thereby measuring the very outer surface hardness. The resulting surface hardness versus the hardening depth was shown in Figure 7A for set A, and in Figure 7B for set B. Test set A show a clear correlation between the surface hardness and hardening depth showing an increasing surface hardness for low hardening depths. This further gives evidence to the correlation between the deep hardening depth and the more superficial BN signal. 

The microstructure for the different samples was also evaluated on polished cross sections. Examples for test set A are seen in Figure 8. The martensitic structure changed with the different induction hardening parameters where the martensitic needles gradually became sharper and coarser with hardening depths (HD). The amount of bainite, also known as troostite, gradually increased as well with increasing hardening depths, observed as small dark spots in the microstructure.

Apart from the great difference observed for the microstructure in the surface, the different samples show more or less sharp microstructural gradients in the transition zone i.e., between the hardened zone and core material. The sharpness of these gradients appears to correlate well to the surface hardness. In Figure 9, two extremes from test set A are shown at different depths: (1) the surface, (2) the effective hardness depth (EHD) and (3) the total hardness depth (THD), which is defined as the depth where the hardness of the core material is reached as defined in SAE J423 [20]. The microstructure clearly differed between the two samples, especially in the transition zones. The left sample with lower EHD showed a slightly lower surface hardness due to presence of retained austenite in the surface, which rapidly changed into a mixture of martensite and ferrite in the transition zone. At the total hardening depth, a sharp transition between different microstructural features was observed. The sample with higher EHD showed fully martensitic surface but a much more gradual transition between different microstructures in the transition zone and no sharp transition at the THD. 

## 4. Analysis and Predictive Modeling

As seen earlier, several NDC-parameters showed a strong correlation with the hardening depth. Further studies of each of these parameter’s relationship to the hardening depth are shown in Figure 10. This highlights the bi-variate correlation of hardening depth versus surface hardness, residual stress, and BN-characteristics for the two test sets. Two possible outliers are statistically identified, explained below (Section 4.1), and marked with stars (*). These are the two deepest hardened samples from both test sets, also identified in Figure 6: the 7 mm sample for set A and the 4.3 mm sample for test set B. These samples also had an elevated hardness at the core of the samples, compared to the samples with a shallower hardening. It is assumed that this influences the RS and BN surface measurements, respectively.

The color-coded matrix of correlations in Figure 11 reveals a heavy bi-variate correlation between the predictors themselves and with the hardening depth. This fact prevents using an ordinary linear regression approach, which assumes predictor independence. The parallel plot in Figure 12 shows the multivariate pattern of the hardness depth characteristics stratified on respective test set. The pattern for both sets is similar and shifts from one pattern for shallow hardening depths and another pattern at deeper. It seems that lack of some signals in combination with others contain hardening depth information of value. This suggests using principal component analysis to find a lower number of independent latent factors that may be used for prediction modeling.

The statistical modeling analysis, using a PCA approach in this work, was targeted to develop prediction models robust for interpolation of the hardening depths in-between measured samples, which otherwise is risky using correlated predictors. The model development consists of three refinement steps:(i)All NDC predictors in a model using ***principal components*** (PC), assuming laboratory set-up independence;(ii)All NDC predictor in a model using ***principal component and principal component interaction*** with Tests sets, assuming laboratory set-up dependence;(iii)Modeling based on minimal set of NDC using BN predictors only, to reduce the monitoring demand.

### 4.1. Prinicple Component Analysis (PCA) of Predictor Correlation

PCA is a technique to reduce correlated variables into a smaller set of independent factors that can be used for ordinary linear regression analysis [21]. Each observation consisted of four + four RS- and BN-characteristics that were projected on the plane defined by the principal components and represented by pairs of PC1- and PC2-coordinates. The plane defined by the two orthogonal principal components capture 87.8% of the variation and removes the correlation between the four + four original characteristics. In Figure 13 all observations are from the test sets projected on the PC1/PC2 plane, test set A (black •), and test set B (red x), respectively. The outliers discussed above marked with a red and a black star were identified with a T2-control chart on the residuals, using the default settings of JMP Pro 16.2. The outliers were kept in the analysis, even though they might not fully represent the same physical phenomena as the other, due to the elevated core hardness. However, they provided some useful information or lack of information that in itself was useful for the elevated understanding. The principal components were themselves linear combinations of the base parameters and represented the main and second most important axis of the variation. These can be interpreted as two perpendicular axes in rotated coordinate system. The coordinates for each sample in the rotated coordinate system were calculated using Equations (3) and (4).
(3)$$\begin{array}{cc} & PC1\;\left(BN+RS\right)=-0.00154\times RS\;Ax-0.00242\times RS\;Tang+0.307\;\times \\ & RS\;FWHM\;Ax+0.308\times RS\;FWHM\;Tang-0.0047\times BN\;RMS+0.127\;\times \\ & BN\;Peak\;Pos-0.236\times BN\;FWHM+1.689\times MVSS\left(\frac{200Hz}{20Hz}\right)-1.325 \end{array}$$
(4)$$\begin{array}{cc} & PC2\;\left(BN+RS\right)=0.00068\times RS\;Ax-0.00179\times RS\;Tang+0.0058\;\times \\ & RS\;FWHM\;Ax+0.026\times RS\;FWHM\;Tang-0.00656\times BN\;RMS+0.109\;\times \\ & BN\;Peak\;Pos+0.590\times BN\;FWHM+1.795\times MVSS\left(\frac{200Hz}{20Hz}\right)-15.104 \end{array}$$

### 4.2. Ordinary Multi-Parameter Linear Regression of Hardening Depth (HD) Using Principal Components—Model i

To test if the principal components based on RS- and BN-measurements were sufficient to predict the hardening depths, a second-order response surface model was created based on PC1 and PC2, fitted to the hardening depth data. A second-order linear model consists of main-effects, two-factor interactions, and quadratic terms. The latter was needed to describe the response surface curvature. The predictive modeling approach, fitting a model to data, was built on three general guiding principles in search for interactions described in [22]: (1) the hierarchy principle—the higher order the less likely the interaction will explain the variation, (2) effect sparsity—only a fraction of the possible effects truly explains the variation, and (3) heredity principle—if an interaction is active, most likely will its parental effects also be. In this case, it resulted in a good prediction model, Equation (5), which captured 86% of the variation in hardening depth (Figure 14A):(5)$$\mathrm{HD}=2.761-0.198\times \mathrm{PC}1+0.190\times \mathrm{PC}2+0.056\times \mathrm{PC}1^{2}+0.127\times \mathrm{PC}1\times \mathrm{PC}2$$

However, the model does not capture all information hidden in the data, when stratifying on Test sets (Figure 14B). The model exaggerates the hardening depth on the test set B samples. This pattern is also indicated in Figure 13, where the test set B samples mainly show negative PC2 coordinates.

### 4.3. Modeling Including Test Sets and Principal Component Interaction—Model ii

The result in the former section implies that it is not sufficient to fully rely on NDC alone to predict the hardening depth, independently of set-up. PC1 captured most of the variation in hardening depth (76%) independently of test sets, that is if there was no separation horizontally between test set A and B, as seen in Figure 13. The general trend was captured, but the precision of prediction model probably increased if it is calibrated with respect to the actual hardening set-up, including sample geometries and application specific alignment. 

Adding test sets as a discrete variable, a stepwise regression suggested a best sub-set model based on both PC1 and PC2 and test sets parameter that explained 90% of the variation in hardening depth. The actual by predicted plots in Figure 15 showed the result of the improved model.

The models predicting the hardening depth in each test set, respectively, were presented in Equations (6) and (7). The interpretation of the coefficient adjusted depending on the test set was that the PC captured the general behavior of the relation between NDC and hardening depths in the same manor for both test sets but needed to be adjusted slightly depending on the application.
(6)$$\begin{array}{cc} {\mathrm{HD}}_{\mathrm{full}}\left(\mathrm{Set}\;A\right)= & 2.624-0.185-\left(0.257+0.175\right)\times \mathrm{PC}1+\left(0.355+0.312\right)\times \mathrm{PC}2\;+\\ & 0.031\times \mathrm{PC}1^{2} \end{array}$$
(7)$$\begin{array}{cc} {\mathrm{HD}}_{\mathrm{full}}\left(\mathrm{Set}\;B\right)= & 2.624+0.185-(0.257-0.175\times \mathrm{PC}1+\left(0.355-0.312\right)\times \mathrm{PC}2\;+\\ & 0.031\times \mathrm{PC}1^{2} \end{array}$$

### 4.4. Residual Analysis for Model Comparison

Analysis of the residuals in Figure 16 revealed a remaining pattern when stratified on test sets. This is an indication of remaining information in the data, Figure 16A, whereas there was no such pattern in the second model when adding test sets and interaction between test sets principal component interaction to the model. This suggests that the second model utilized the revived data, better indicated by the higher goodness of fit score of the second model.

### 4.5. Modeling Including Test Set and Principal Componets Based on BN-Characteristics—Model iii

To reduce the need for monitoring, it would be interesting to use BN-characteristics only. A second set of principal components using the four BN characteristics only is shown in Equations (8) and (9):(8)$$\mathrm{PC}1\;\left(\mathrm{BN}\right)=-0.007\times \mathrm{BN}-\mathrm{RMS}\;+0.193\times \mathrm{BN}\;\mathrm{Peak}\;\mathrm{pos}\;-0.364\;\times \;\;\mathrm{BN}\;\mathrm{FWHM}\;+2.657\times MVSS\left(\frac{200Hz}{20Hz}\right)+6.332$$
(9)$$\mathrm{PC}2\;\left(\mathrm{BN}\right)=-0.007\times \mathrm{BN}-\mathrm{RMS}\;+0.146\times \mathrm{BN}\;\mathrm{Peak}\;\mathrm{pos}+0.628\;\times \;\;\mathrm{BN}\;\mathrm{FWHM}-1.394\times MVSS\left(\frac{200Hz}{20Hz}\right)-16.576$$

Figure 17 shows the performance of the regression model stratified on test sets. It captured the variation in the data very well (R^2^ = 0.92). The models predicting hardening depth based in BN characteristics scaled for the different test sets are shown in Equations (10) and (11):(10)$${\mathrm{HD}}_{\mathrm{BN}}\left(\mathrm{Set}\;A\right)=2.815-0.157-\left(0.502+0.259\right)\times \mathrm{PC}1\left(\mathrm{BN}\right)+\;\left(0.234+0.314\right)\times \mathrm{PC}2\left(\mathrm{BN}\right)$$
(11)$${\mathrm{HD}}_{\mathrm{BN}}\left(\mathrm{Set}\;B\right)=2.815+0.157-\left(0.502-0.259\right)\times \mathrm{PC}1\left(\mathrm{BN}\right)-\;\left(0.234-0.314\right)\times \mathrm{PC}2\left(\mathrm{BN}\right)$$

## 5. Discussion

This work demonstrated the potential using NDC methods for monitoring of surface hardening processes that had the potential to reducing the need of process stand-still after resetting during destructive verification. It was further shown that there may exist some individual differences originating of the hardening process itself, comparison between set A and B, which requires individual models to be built for different process setups. The difference between the two sets could further be explained by the variation and differences in quenching, the efficiency of the quenching shower, as well as of different induction coils that may heat the parts differently, including the individual coupling distances. 

The destructive evaluation of the samples was important in order to understand the correlation with the BN signal. The correlation between the BN signal and HD was believed to be a combination of differences in the microstructure as well as in the surface hardness. The destructive analysis showed that the sharpness of the transition zone differed greatly between the samples with different HD as well as the very outer surface hardness. This explains why there exists a correlation between deep hardening depth of several millimetres and the shallow signal of Barkhausen noise of a few hundreds of penetration depth. 

The statistical analysis revealed that modeling directly using the NDC is difficult due to heavy correlation among the predictors that prevents ordinary linear regression. The two-step procedure consisted of two steps: first, reducing the number of correlated predictors to a few orthogonal principal components. This represents the main physical behavior of the predictors. In this case, the set included four residual stress characteristics and the four BN characteristics. One underlying advantage using this approach was that the principal components captured the combined pattern of metrics to a larger extent, and even a lack of signal is useful information. For example, when the BN-signal became weaker but not the RS signal, this was also a characteristic pattern that was useful. The second step was to use the orthogonal principal components as predictors when fitting a predictive model to the data. In this case, a second-order response surface model was sufficient. 

The predictive modeling toolbox contains many different kinds of modeling techniques, from ordinary linear regression to more advanced non-linear techniques, such as general regression or neural networks, etc. It is important to remember, however, that the purpose with the models using this approach is not to explain the physics behind. It only makes it possible to handle the physics and to prioritize the likelihood of the importance of the parameters in a certain case, which is sometimes invaluable when there are lots of characterization and process control parameters involved. After the model building phase, the models can be re-transformed to the natural parameter space and it is possible to build the foundation for monitoring systems in the next phase. An alternative approach would be to use partial least square regression, PLS, in this case, which finds latent factors that capture the variation of both predictors and responses in one step, connecting them directly. The gain is somewhat simpler model building. One practical difference between the approaches is that the principal components above can be used to model other responses without losing the general connection to the physical behavior of the predictors. In PLS, the latent factors connect them. Both methods may be used to explore if the hardening depths are related to the NDC characteristics within the range tested. The analysis further revealed that the precision of the predictions further can be improved if the model is calibrated relative the specific application regarding sample shape and hardening application geometry. This implies that the even though the general pattern of how combination of characteristics react to variations of the hardening depth is the same, the variations of sample geometry and induction hardening equipment set-up influence the scaling of the coefficients, which make them case-dependent. Understanding exactly how such variations influence, and if and to what extent and how the monitoring further can be simplified, remains to be explored.

## 6. Conclusions

The concluding results from depth measurements and statistical modeling of induction hardening samples show that:The different Barkhausen noise parameters, RMS/FWHM and MVSS(200 Hz/20 Hz), correlate well to hardening depths down to 4.5 mm;The surface hardness and hardening depth correlate, which explains the Barkhausen noise signal sensitivity to the several-millimetres-deep correlation to the hardening depth;It is possible to predict the hardening depth using principal components based on all or a reduced set of BN and RS characteristics;The test set dependence indicate that the BN and RS characteristics capture subtle differences of the hardening result hidden in the microstructure whether it depends on variations of the base material or slightly different heating and cooling efficiency in the elaborative set-up;The surface hardness and hardening depth are correlated, which explains the correlation between the hardening depth and the Barkhausen noise signal.

## Figures and Tables

**Figure 1 micromachines-14-00097-f001:**
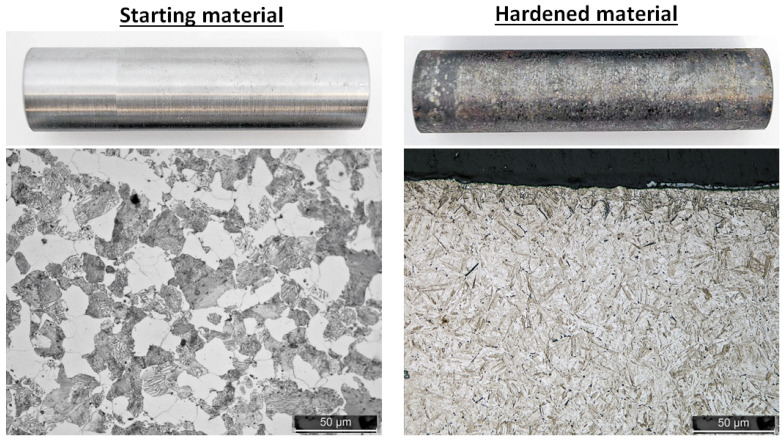
Overview of surfaces and microstructures before and after induction hardening.

**Figure 2 micromachines-14-00097-f002:**
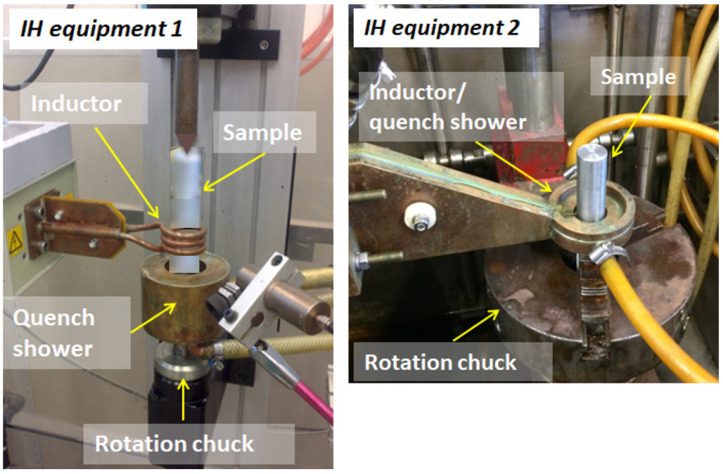
Overview of the induction heat treatment equipment used to harden the samples.

**Figure 3 micromachines-14-00097-f003:**
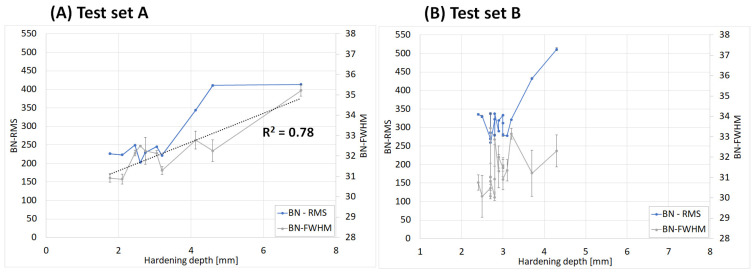
Barkhausen noise RMS and FWHM versus the hardening depth, (**A**) set A and (**B**) set B.

**Figure 4 micromachines-14-00097-f004:**
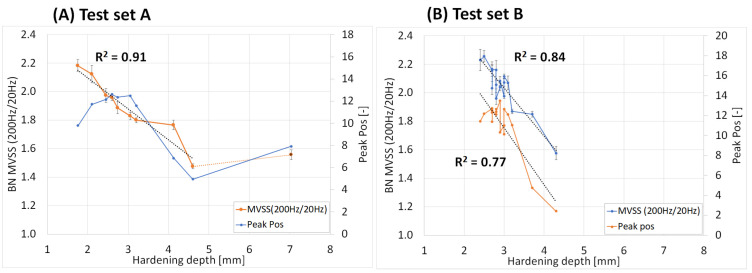
Peak position and MVSS (200 Hz/20 Hz) versus hardening depth, (**A**) set A and (**B**) set B.

**Figure 5 micromachines-14-00097-f005:**
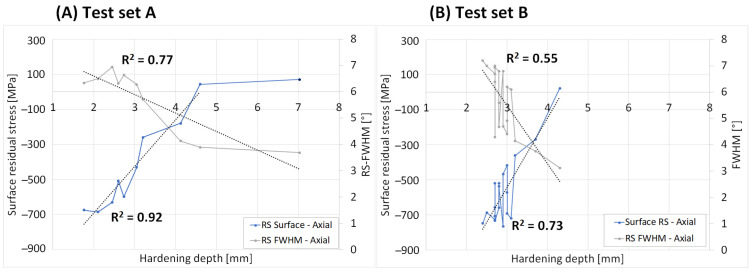
Surface residuals stresses versus the hardening depth for (**A**) set A and (**B**) set B.

**Figure 6 micromachines-14-00097-f006:**
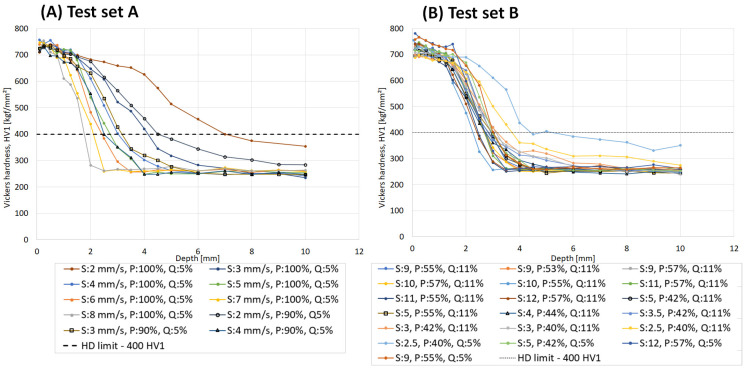
Hardness profiles, Vickers (HV1), for (**A**) test set A and (**B**) test set B, induction hardened samples with different scanning speed (S), power (P), and quenchant concentration (Q).

**Figure 7 micromachines-14-00097-f007:**
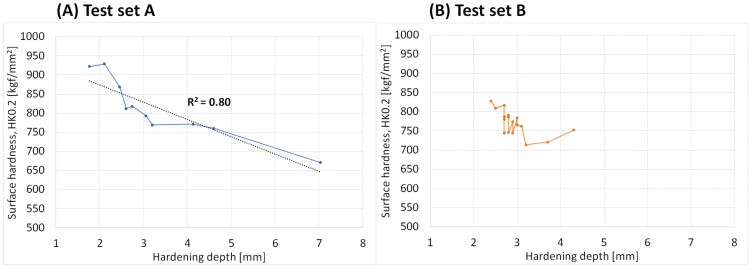
Surface hardness, Knoop (HK0.2), versus the hardness depth of the induction hardened samples from (**A**) test set A and (**B**) test set B.

**Figure 8 micromachines-14-00097-f008:**
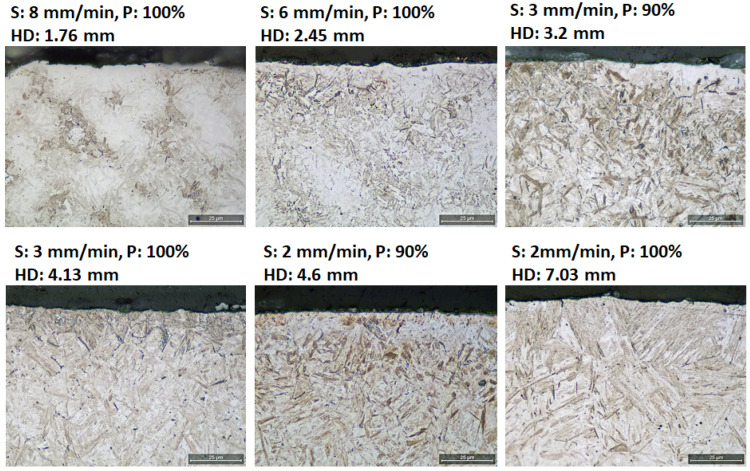
Micrographs of selected samples, induction heat-treated with different scanning speed (S) and power (P), with gradually increasing hardening depth (HD).

**Figure 9 micromachines-14-00097-f009:**
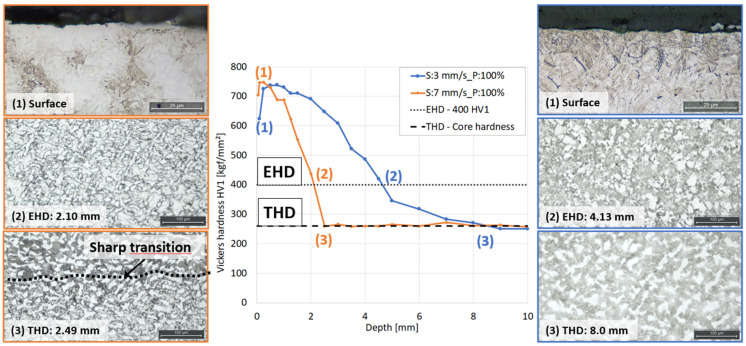
Micrographs at different depths (EHD: effective hardness depth, THD: total hardness depth) for left: sample with scanning speed (S) of: 7 mm/s and right: sample S: 3 mm/s.

**Figure 10 micromachines-14-00097-f010:**
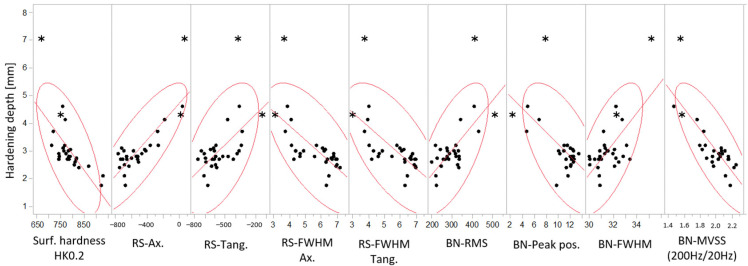
Varying correlation between actual hardness depths at 400 HV versus different predictors, surface hardness, RS, and BN monitoring characteristics. Possible outliers marked with (*).

**Figure 11 micromachines-14-00097-f011:**
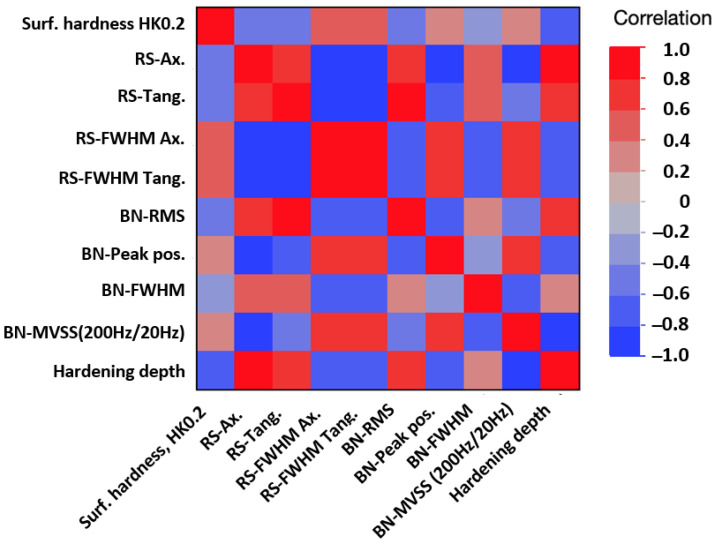
Heavy pairwise correlation between the response and predictors, both within and between residual stress (RS) and Barkhausen noise (BN) characteristics.

**Figure 12 micromachines-14-00097-f012:**
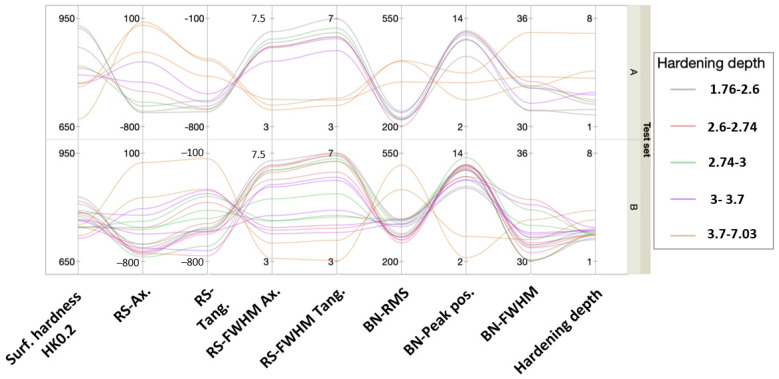
Parallel plot of the hardness depth characteristics stratified on test set.

**Figure 13 micromachines-14-00097-f013:**
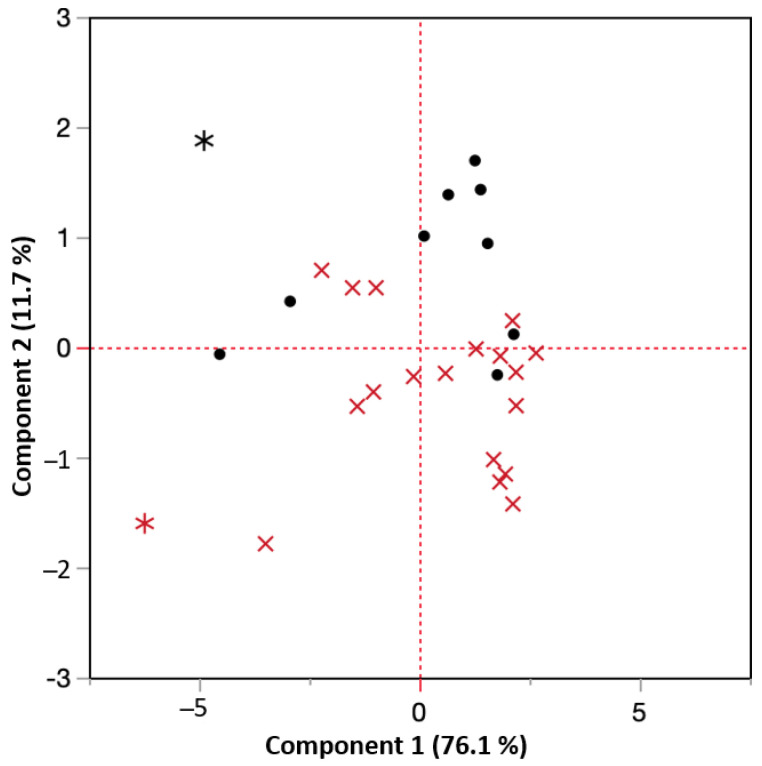
Score plot of all observations projected on the principal component plane: Test set A (•) and test set B (x). Outliers, identified using default T2-statistics in JMP, are marked with stars (*).

**Figure 14 micromachines-14-00097-f014:**
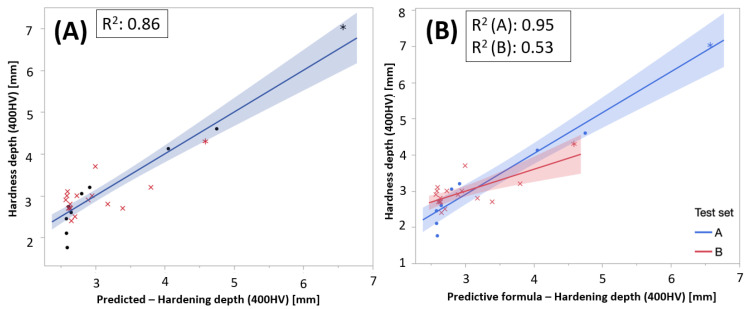
(**A**) Actual vs. predicted plot for the model (i), based on principal components only and (**B**) stratified on test sets.

**Figure 15 micromachines-14-00097-f015:**
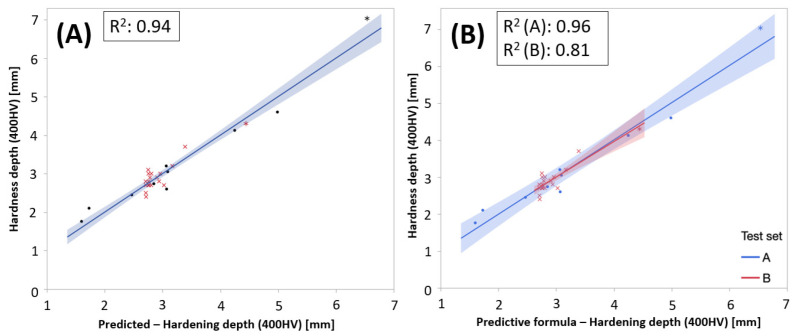
(**A**) Actual vs. predicted plot for the developed model and (**B**) stratified on test sets.

**Figure 16 micromachines-14-00097-f016:**
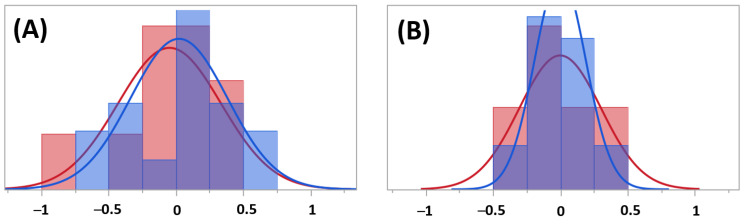
The residuals, stratified on test sets (red for set (**A**) and blue for set (**B**)), from (**A**) modeling hardening depths using principal components only (model i) and (**B**) the improved model (ii).

**Figure 17 micromachines-14-00097-f017:**
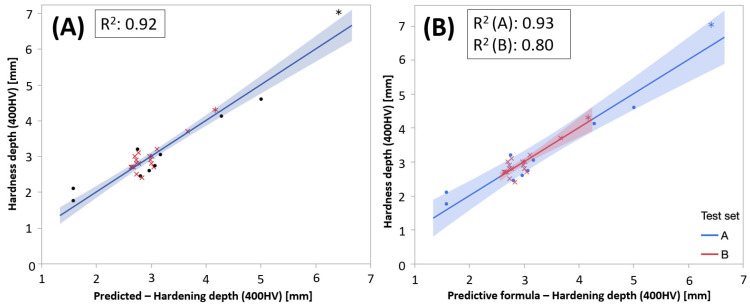
Actual by predicted using the third minimal model based on only BN-parameters.

**Table 1 micromachines-14-00097-t001:** Induction hardening settings of the two test sets, set A and B.

Sample Set	Scanning Speed [mm/s]	Power [% of Full Power]	Full Power [kW]	Quenchant Concentration [%]
A	2–8	90/100	50	5
B	2.5–12	42–55	150	5, 11

## Data Availability

Not applicable.

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
