# Peer review of "Predictive Modeling of Induction-Hardened Depth Based on the Barkhausen Noise Signal"

_micromachines, 2022, doi:10.3390/mi14010097_

Round 1
Reviewer 1 Report
1.The surfaces residual stresses show a very strong 189 linear correlation to HD for set A, as shown by the high R2 values of linear trendlines if 190 the 7 mm sample is considered as an outlier. Is it good to say very strong correlation for R2 = 0.91
2. What is the importance of figure 7. surface hardness vs hardness depth. can u please elaborate the significance of this figure.
3. Please include the input parameter setting for BN analysis to improve readability.
Author Response
1.The surfaces residual stresses show a very strong 189 linear correlation to HD for set A, as shown by the high R2 values of linear trendlines if 190 the 7 mm sample is considered as an outlier. Is it good to say very strong correlation for R2 = 0.91
Response: Agree, this is a valid comment. We have removed the wording strong, see line: 206.
- What is the importance of figure 7. surface hardness vs hardness depth. can u please elaborate the significance of this figure.
Response: Thanks for the comment, We have clarified this in the text, see line: 226-228 and 232-233. This Figure is a very important result since it explains why we get the correlation between the deep hardening depth and the superficial BN signal. Normally, hardening depths is measured using HV1, but this is a too rough method to study the very outer surface hardness, hence Knoop needs to be used and then the correlation is obvious.
- Please include the input parameter setting for BN analysis to improve readability.
Response: The settings for the burts analysis have been added to the text, see line 162-164. All BN other measurement settings are mentioned in the experimental section.
Reviewer 2 Report
1. Please add the explanation for the principle of Barkhausen noise.
2. The literatures are not fully reviewedand the explanation of the relationship between Barkhausen noise and hardening depth should be included.
3. The relationship between the Magnetic Voltage Sweep Slope and Barkhausen noise should be explained deeply.
4. The symbols in Equations (1) and (2) are not fully explained.
5. The last paragraph of Introduction section should give an outline of the study.
6. In the results section, the comparison between sample A and B is not discussed enough.
7. References should be added to the definition of effective hardening depth (EHD).
8. What is the propose of destructive evaluation in this study?
9. an deeper model explanation and more results discussion are recommended .
10. the grammar of manuscript should be checked carefully.

Author Response
- Please add the explanation for the principle of Barkhausen noise.
Response: We have added text and two references explaining BN further, see line: 47-50.
- The literatures are not fully reviewed and the explanation of the relationship between Barkhausen noise and hardening depth should be included.
Response: The relationship between the hardening depth and BN signal is not fully known in the correct literature as mentioned in the manuscript. This has been one target in this work, and we show that there is a correlation between the surface hardness and the hardening depth which explains why the shallow BN signal could measure the deep HD. Further, in the existing literature limits measurements of the HD to 3 mm while in this work we expand the depths to 7 mm.
- The relationship between the Magnetic Voltage Sweep Slope and Barkhausen noise should be explained deeply.
Response: The text has been adjusted accordingly and hopefully clarified the MVSS signal, this can be seen on line: 62-68.
- The symbols in Equations (1) and (2) are not fully explained.
Response: This has been adjusted in the manuscript, on line: 75, 84 and 86-87.
- The last paragraph of Introduction section should give an outline of the study.
Response: Good idea, this has been added to the manuscript on line: 121-126.
- In the results section, the comparison between sample A and B is not discussed enough.
Response: In the discussion section this comparison has been added, see line: 414-420.
- References should be added to the definition of effective hardening depth(EHD).
Response: This has been adjusted to the manuscript by adding a reference, see line: 215 and 253-254, reference 21.
- What is the propose of destructive evaluation in this study?
Response: The destructive evaluation is essential since we need to evaluate the response from the heat treatment and use that data to correlate and build the predictive model. However, the target of this work is to completely remove that step in production today, which implies the strength of the success of this work.
- an deeper model explanation and more results discussion are recommended.
Response: Agree, we have expanded the discussion section accordingly.
- the grammar of manuscript should be checked carefully.
Response: The complete manuscript has been reviewed again.